# Experience of Pro-Poor Tourism (PPT) in China: A Sustainable Livelihood Perspective

Zhaoguo Wang and Fengli Dong *

College of Economic and Management, Shenyang Agricultural University, Shenyang 110065, China
* Correspondence: dongfengli@syau.edu.cn

**Abstract:** Pro-poor tourism is a powerful tool in China's poverty alleviation strategy, helping the achievement of Sustainable Development Goal 1, no poverty. Thus, the Chinese experience in tourism poverty reduction could be instructive. Considering the dominant role of the government in PPT, this study examines the government's PPT scheme within a sustainable livelihood framework, uncovering the usefulness of PPT in poverty alleviation. With thematic analysis, 18 cases are systematically coded, and several findings are discovered. Rural development is co-evolutionary with PPT, while livelihood capitals change correspondingly. Specifically, human capital is mentioned as the top priority, followed by economic capital, institutional capital, social capital, and natural capital. Analysis of the cases indicates that livelihood capital comprises multiple themes, and a variety of livelihood strategies are applied conditionally. Additionally, livelihood outcomes are in accord with the criteria of Beautiful Village, characterized by good quality of life. In summary, the success of the PPT in China is a comprehensive project, contributed to by a government-led model, a well-organized community system, effective community participation, and whole-of-society synergy. The study demonstrates that a paradigm shift has been seen in China's PPT model and sheds light on tourism development in poverty alleviation globally.

**Keywords:** pro-poor tourism; government narratives; thematic analysis; China

## 1. Introduction

Poverty eradication is the first and foremost objective among Sustainable Development Goals (SDGs) [1]. The tourism industry is labor-intensive, with high industrial linkages and low industry thresholds [2–6], therefore, the key to alleviating these challenges lies in providing alternative livelihood strategies, reducing the vulnerability of poor areas, and enhancing their well-being [7–9]. Given the trickle-down effects of reducing poverty [2,10], tourism has acted as the most potent tool in achieving the SDG1, No Poverty [11,12].

The contributions of the tourism industry to poverty alleviation were first noted in the 1970s [13]. After that, the UK's Department for International Development (DFID) proposed the term "*Pro-poor tourism*" (PPT) in 1999 [14,15] and various PPT programs have emerged since this time [6]. PPT is an overall approach designed to unlock opportunities for the poor instead of benefitting a specific tourism product or sector [16], an approach which arises from a belief that tourism can and should contribute to pro-poor economic growth.

However, PPT is contested, as well as the tourism-poverty link [17]. A debate always exists concerning the effectiveness of PPT in poverty alleviation [9,11,18]. The tourism-poverty link is contradictory, and it is found that tourism development has no systematic effects on the poor [19]. In Vietnam, results indicated that most participants considered tourism to be a contributor to poverty alleviation in Vietnam [20]. Moreover, the adverse effects of tourism development remain unsolved in some PPT cases [21,22]. Because of the contradictory impacts of tourism on poverty, capitalizing on these advantages for the poor and reducing negative impacts are challenging for PPT.

Within the PPT framework, alleviating poverty and helping the poor are set as targets [23]. Under this poverty-centered principle, PPT reduces poverty by increasing net

benefits for the poor or directing profits back into the community [4,15,23]. Most empirical studies of PPT are conducted in developing or undeveloped regions, where tourism revenue matters for community development [11,19]. In many cases, the poor benefit less than the wealthy and powerful [5]. Thus, attention to equity is essential to genuinely achieve the alleviation of poverty [24]. Moreover, meaningful community participation, relationship building, and ethical decision-making are necessary to skew benefits in favor of the poorest. By transmitting social, environmental, cultural, and economic benefits to the deprived [11,12,25], PPT improves community development and assists in accomplishing SDG1. The implementation of PPT must critically consider how to reap significant and long-term benefits for the poor and fringe communities [17].

Additionally, it is no easy task to adopt PPT as a new livelihood due to PPT's weak sustainability [23]. The poor are usually less skilled in the tourism industry [26] and the low level of human capital aggravates their deprivation [27]. In addition, the absence of effective cooperation between stakeholders hinders the poverty alleviation efforts of tourism development [15]. The poor are excluded from the thriving tourism industry because of their lack of pollical agency [28]. Therefore, government intervention is required to overcome the inherent deficits of tourism development in rural areas [29]. However, neoliberal policies advocated by western governments prevent intervention targeting equity within tourism, which restricts 'pro-poor tourism' strategies [24].

In contrast, the positive impact of tourism on poverty alleviation is encouraged by the Chinese government [3]. China has achieved marvelous success in poverty reduction under the *Poverty Alleviation Tourism Policy* [30,31]. The Ministry of Culture and Tourism of China (MCTC) has selected 22,651 villages as national PPT villages across the Chinese mainland [32]. PPT has played a significant and favorable role in China's poverty alleviation efforts, contributing to about 30% of the country's poverty elimination success; as of 2020, nearly 12 million poor people have been lifted out of extreme poverty [33]. The contribution of PPT to poverty alleviation within China offers valuable insights compared to various parallel situations globally, and the scrutiny of policy narratives is critical in determining the comprehensive effects of PPT [34–36].

Research on PPT in China has been conducted to explore relevant experiences and characteristics [3,15,22,30,37,38], providing a glance at the secrets of successful PPT in China. However, the present studies make it hard to see the whole picture of PPT in China. Additionally, dialogue with the western discourse on PPT is absent. Consequently, the paper tries to answer the question: why has PPT in China succeeded, and is it practiced in a sustainable way? A sustainable livelihood framework (SLF) is applied to reveal the secrets of PPT's success in rural China. We also contribute to the tourism-poverty literature in many ways. Firstly, we analyze the systematic transformation of livelihoods t generated by PPT policies in China. Secondly, our results extend the knowledge of PPT under a government-led model using official evidence. The remaining sections of the paper are structured as follows. Section 2 presents a review of the literature. Section 3 describes the data and methodology used. Section 4 reports the results. Section 5 presents the discussions. Section 6 summarizes our conclusions and their implications.

## 2. Literature Review

### 2.1. Livelihood Effects of PPT in China

A livelihood is a way of gaining a living, which is not just about income and employment, but also about finding diverse strategies for making a living [39,40]. Livelihood improvement is the critical target in poverty reduction, and sustainable livelihood is crucial to meet the standards of SDG1 [15]. Tourism is a valuable tool for livelihood diversification and community development [8,41] and tourism as a livelihood has been an important topic in tourism research [21,42]. Plenty of tourism products benefit locals and support the livelihood of the rural community, such as tourism homestays [40], agroecological tourism [43], and food tourism [7]. In addition, all the studies mentioned above indi-

cate that sustainable livelihood is the most helpful instrument for analyzing the poverty reduction effects of tourism development [26].

China has recognized the importance of tourism development in poverty alleviation, and the government plays a decisive role in the PPT [3,12,44]. In 1996, the former China National Tourism Administration (CNTA) introduced poverty alleviation tourism as a national policy. The term "fu pin lv you" was introduced to address PPT [38], which acts as a specific type of rural tourism [45]. In 2000, the CNTA proposed the first National Tourism for Poverty Alleviation Pilot Zone, and CNTA started to advocate and promote rural tourism demonstration examples in 2006 [31,46]. The form that PPT takes is varied, including "*nong jia le*" tourism, folk-custom tourism, rural eco-tourism, agrotourism, leisure farm tourism, etc. Additionally, six operational models had been identified under the cooperation with corporations, communities, government, farmer cooperatives, or individuals [46]. More than 800 billion yuan was generated from 3 billion tourist visits to rural areas in 2018 [47]. As a result, tourism development increases the livelihood of the poor [29]. Generally, tourism transforms the economy, society, and environment of rural areas in China and the community's livelihoods. Nevertheless, considering institutional, organizational, and location factors, some researchers suggest that high dependence on tourism may reduce the sustainability of community livelihoods [21]. Thus, studies should thoroughly probe the practice of PPT in China to realize sustainable livelihood and poverty elimination.

### 2.2. Sustainable Livelihood Framework

The Sustainable Livelihood Framework has been widely used to understand the multi-dimensional aspects of poverty and provide solutions [39,48]. The SLF was first put forward in the 1980s [4,49]. Despite the lack of uniformity of the SLF, the DFID scheme is a very popular way of organizing the complex issues surrounding poverty [14]. It is worth pointing out that SLF needs to adjust to local circumstances and priorities. Based on the DFID scheme, several SLFs have been reconstructed and proposed to adapt to various contexts [4,21]. Because institutional and pollical factors have a profound role in China's economic and social development [50], institutional capital has been applied well in China [18]. Thus, this paper applied the structure of livelihoods and capital as comprising human capital, social capital, natural capital, economic capital, and institutional capital [4].

Generally, the SLF consists of five key elements namely, context, conditions and trends, livelihood resources (capitals), institutional processes and organizational structures, livelihood strategies, and sustainable livelihood outcomes [4,40,51,52]. To reveal the success path of PPT in China, the transformation of livelihood resources (capital) and livelihood outcomes are focused on and explored, while other components of SLF are outside the scope of this paper. Additionally, the rural community is discussed instead of families or individuals to capture a full view of sustainable livelihood under the PPT.

### 3. Data and Methods

### 3.1. Data Collection

The Chinese government aims to advocate for the success of PPT in rural areas. In 2021, the MCTC launched the selection of PPT exemplars in rural areas, advocating the marvelous achievement of China's efforts and the experience of China's poverty alleviation through tourism policy. One hundred cases have been selected and classified into six types: industry integration-oriented, entrepreneurship and employment-oriented, cultural inheritance-oriented, ecology protection-oriented, rural governance-oriented, and innovation and upgrading-oriented. Thereinto, industries integration-oriented cases emphasize the agglomeration effects of tourism development by establishing an integrated value chain and providing alternative livelihoods alongside the dominant industry. The multi-livelihood strategy is well applied in this type of case, offering robust empirical materials to investigate the role of PPT in poverty reduction.

On 1 December 2021, qualitative data were obtained from the official website of the MCTC. Altogether, 24 cases are gathered in the integration-oriented type. The study

focuses on PPT conducted by rural communities rather than other organizations. Then, 18 cases were examined (Table 1). All the cases are structured uniformly, comprising basic information, poverty alleviation efforts, methodologies, enlightenment, and prospects. A total of 54,333 words provides a sound description of the role of PPT in rural poverty reduction. They are characterized by varied poverty rates before the PPT policy was in place (from 5.53% to 90.04%) and are distributed in 16 provinces randomly. Under the PPT model, all exemplar cases have prospered in terms of livelihood and industry development.

**Table 1.** Integration-oriented cases of PPT in China.

| Case | Code | Text Count | Poverty Rate (%) | Province |
|------|------|-----------|------------------|----------|
| *Anqing village* | R1 | 3814 | 5.53 | Guangdong |
| *Bamou village* | R2 | 2739 | 61.95 | Guangxi |
| *Baishuidong village* | R3 | 3120 | 12.29 | Hunan |
| *Danankeng village* | R4 | 2936 | 16.51 | Anhui |
| *Deji village* | R5 | 3041 | 90.04 | Qinghai |
| *Dingzhuang village* | R6 | 4104 | 6.55 | Jiangsu |
| *Gangsha village* | R7 | 2366 | 23.17 | Xizang |
| *Haohuahong village* | R8 | 2746 | 10.87 | Guizhou |
| *Huaguo village* | R9 | 2867 | 34.02 | Shannxi |
| *Huaguoshan village* | R10 | 3090 | 33.02 | Hebei |
| *Lianhuadong village* | R11 | 2504 | 17.9 | Chongqing |
| *Longwangba village* | R12 | 3007 | 51.49 | Ningxia |
| *Sanjia village* | R13 | 2661 | 25.4 | Hunan |
| *shepeng village* | R14 | 4058 | 25.46 | Guizhou |
| *Taigedou village* | R15 | 2522 | 9.6 | Inner Mongolia |
| *Tantou village* | R16 | 2802 | 14.46 | Jiangxi |
| *Yanwoyuan village* | R17 | 3247 | 35 | Hubei |
| *Zhonghaoyu village* | R18 | 2709 | 60.18 | Shandong |

*3.2. Thematic Analysis*

Thematic analysis (TA), an inductive qualitative method, provides a new way of probing the empirical world by identifying themes [53–56]. Additionally, TA potentially reduces voluminous data into clearly articulated thematic statements and is beneficial when a scholar seeks to depict a detailed picture of a focused phenomenon [55]. Therefore, TA is chosen as a primary method to describe the studied phenomenon and contributes to the emergence of new theoretical directions of the empirical world. The framework of TA proposed by Braun and Clarke is accepted for its validity in qualitative tourism research [56]. Figure 1 shows six steps with partial adjustments. It is noted that the TA process would need to be recursive to identify possible themes. In brief, TA is utilized to code the cases under the guidance of the SLF, revealing the effects of PPT on livelihood capitals and outcomes. The process is performed with the software of NVivo 12.

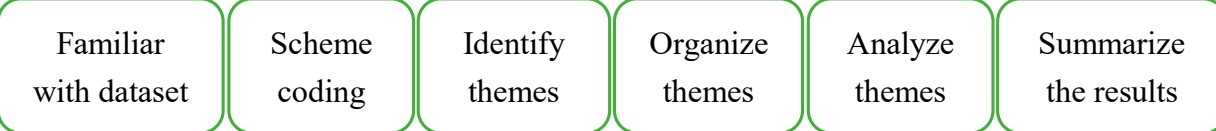

**Figure 1.** Steps of thematic analysis [53–55].

Familiarity with the qualitative dataset is critical for further analysis. Following the SLF, an a priori coding scheme is constructed to focus on the most meaningful information, ensuring the application of the codes is consistent throughout the coding process. Most possible codes are determined after several cycles of coding. A theme-based approach would lead to new insights with each reading of the texts. Table 2 depicts the coding process of the PPT cases, and 1380 reference points and 370 codes are recognized.

**Table 2.** The coding process of PPT cases.

| Case | R1 | R2 | R3 | R4 | R5 | R6 | R7 | R8 | R9 |
|---|---|---|---|---|---|---|---|---|---|
| Code number | 23 | 22 | 21 | 20 | 24 | 22 | 18 | 21 | 19 |
| Reference point | 125 | 62 | 93 | 79 | 87 | 114 | 47 | 102 | 70 |
| **Case** | **R10** | **R11** | **R12** | **R13** | **R14** | **R15** | **R16** | **R17** | **R18** |
| Code number | 21 | 19 | 19 | 20 | 19 | 22 | 19 | 22 | 19 |
| Reference point | 72 | 60 | 69 | 64 | 52 | 59 | 72 | 105 | 48 |

Afterward, basic themes are created from these codes. An organized theme is checked for coherence and consistency by reviewing the data extracts for each code. The SLF guides the analysis of the dataset in a logical, inductive way. The last step is conducted by reviewing and articulating all of the themes of livelihood capitals and outcomes in SLF. They are identified and summarized in Table 3.

**Table 3.** The thematic analysis process with SLF.

| SLF (Times Mentioned) | Themes | Code Category | Case |
|---|---|---|---|
| Economic capital (124) | Industry convergence | Multifunctional agriculture | R1, R6, R9, R12, R18 are excluded |
| | | Poverty alleviation industry | R1, R4, R10 |
| | Tourism development | Rural experience | R1, R3, R5, R9, R12, R15, R16 |
| | | Rural scenery | R4, R5, R6, R8, R10, R11, R12, R14 |
| | | Tourism service | R2, R3, R6, R10, R12, R13, R15, R17 |
| | Community development | Infrastructure promotion | R1, R4, R5, R9, R12 |
| | | Rural landscape promotion | R2, R3, R6, R11, R13, R16, R17 |
| | | Environment promotion | R2, R3, R8, R9, R12, R13, R16, R18 |
| | | Public service promotion | R1, R2, R6, R18 |
| Human capital (134) | Community participation | Tourism employee | R1, R3, R4, R5, R6, R8, R11, R12, R14, R18 |
| | | Non-tourism employee | R1, R3, R4, R6, R8, |
| | | Participation competencies | R1, R8, R9, R10, R11, R13, R14, R16, R17, R18 |
| | Community cohesion | Community self-governance | R1, R2, R4, R7, R12 |
| | | Management capability | R1, R2, R4, R12, R13, R15, R17 are excluded |
| | | Leadership | R5, R10, R11 are excluded |
| | | Cultural construction | R1, R2, R3, R5, R6, R8, R13 |
| Social capital (49) | Social resources | Collaboration networks | R5, R7, R8, R12, R18 are excluded |
| | | Social investment | R7, R10, R11, R14, R15, R17 are excluded |
| Institutional capital (71) | Brand effects | Community honor | R7 is excluded |
| | Institutional effects | poverty alleviation mechanism | All cases are included |
| | | Poverty alleviation model | R12 is excluded |
| Natural capital (37) | Geographical advantage | Tourism resources | R2, R9, R12, R16 are excluded |
| | | Location condition | R3, R4, R8, R17 are excluded |
| Livelihood outcomes (42) | Poverty alleviation | Get rid of poverty | R8, R11, R13, R15, R16 are excluded |
| | Tourism livelihood | Tourism benefits | R1, R4, R9, R14 are excluded |
| | Quality of life | Living conditions | R1, R5, R6, R7, R17, R18 |

## 4. Results

Tourism has been the prevailing means of rural transformation, making profound socio-economic changes and diversifying rural livelihoods [7,41,57], especially when accomplished through PPT [4,58]. Based on the thematic analysis of the PPT cases, the order of each livelihood capital mentioned in the cases is human capital (134) > economic capital (124) > institutional capital (71) > social capital (49) > natural capital (37). Additionally, livelihood outcomes are mentioned 42 times in all cases. As outlined in the government

narratives, livelihood capital and outcomes are valued to varying degrees; they will now be discussed in turn to reveal the transformation caused by the PPT model.

### 4.1. Themes of Human Capital

Human capital represents laborers' overall capacity that enables people to pursue different livelihood strategies, including building skills, knowledge, ability to work, maintaining good health, etc. [4,8,14,40]. Tourism is a labor-intensive industry [3] and low human capital has been the biggest obstacle to the implementation of the PPT [59]. Human capital is thoroughly coded as the core of livelihood capital for improving community livelihood. Code categories are identified and grouped into themes using TA. Community participation and community cohesion are the two themes of human capital, while participation competencies and leadership are each theme's most popular code category. Specific cases would provide additional details.

Community participation is the guarantee of sustainable livelihood. As revealed by Feng and Li [32], the participation of the poor is at the core of PPT. In the case of R8 (*Haohuahong village*), relevant content was coded as tourism employee, non-tourism employee, and participation competencies. Villagers are provided with sustainable livelihood strategies derived from PPT, achieving benefit sharing as employees, entrepreneurs, etc. The village emphasizes the promotion of competent participation. Poor people are trained to adapt to alternative livelihoods, such as traditional craft production and homestay operation, thus increasing their participation quality. The promotion of community participation has been achieved in numerous ways, including free skill training, a robust decision-making process, and investment opportunities [31,60].

Community cohesion is the cause and outcome of sustainable livelihood. Community support is essential for rural development, especially in PPT. In the case of R7 (*Gangsha village*), relevant content was categorized as community self-governance, leadership, and management capacity. As a result of collective decisions, *Gangsha* established *Kangrinboqe* tourism service company under the guidance of the committee and the grassroots organization of the Communist Party of China (CPC). Moreover, skilled and talented candidates are selected as managers to enhance management efficiency. Additionally, community culture construction also acts as a valuable path to enhance cultural identity and community cohesion. For example, in the case of R6 (*Dingzhuang village*), cultural construction is boosted by the touristification of rural heritage and agricultural civilization, and cultural spaces are constructed to foster community cohesion. In short, China's rural areas have benefited from village leader-led or elite-led cooperatives [7,30], and community elites, social organizations, and companies are activated to improve community cohesion [61].

### 4.2. Themes of Economic Capital

Economic capital is indispensable for pursuing any livelihood strategies [40] comprising financial resources and basic infrastructure [4]. In line with the coding process of human capital, this chapter concludes with three themes, namely industry convergence, tourism development, and community development. Multifunctional agriculture, tourism services, and environment promotion are each theme's most popular code categories, respectively.

The general requirements of rural revitalization are "*prosperous industries, livable ecology, civilized rural customs, effective governance, and successful life*" [62]. Thereinto, industrial prosperity is the prerequisite for solving all problems in rural areas. In China, pro-poor industries adopt a variety of forms to take maximum advantage of rural resources, and PPT provides the most valuable model. In the case of R6 (*Dingzhuang village*), relevant content was coded as multifunctional agriculture. The village forges a whole industry chain to gain added value from grapes and their byproducts. Multifunctional agriculture is a tourism-based farm diversification with high added value and is supported by the government [25,43,44]. In contrast, non-tourism industries provide alternative livelihoods as well, and anti-poverty projects are implemented. In the case of R1(*Anqing village*), relevant content is coded as belonging

to the poverty alleviation industry, while photovoltaic power generation, hydropower stations, and chicken raising are primary poverty alleviation measures.

High-quality tourism services are essential to rural tourism, and PPT improves tourism development to meet various demands. In the case of R3 (*Baishuidong village*), relevant content was coded as rural experience and tourism services. Festival activity such as the "*Yellow peach festival*" was launched, making *Baishuidong* an attractive scenic spot. The village also built a skilled education base to serve the students of Shaoyang city, enabling the locals to participate by providing services. *Baishuidong* tries to build an integrated tourist area with a series of tourism services, such as rural homestays, botanical gardens, health care facilities, and fishing gardens. Moreover, touristification of the rurality is the key to attracting visitors [28,43,44], and the rural experience is improved through PPT initiatives. In the case of R10 (*Huaguoshan village*), relevant content was coded as rural scenery. The village is located in the *Huaguoshan* tourism area, famous for its traditional people, Great Wall culture, and mountain scenery. The locals provide abundant tourism services, such as donkey riding. It should be noted that the local government dominates the touristification of the village, and more than 100 million yuan has been invested to promote tourism infrastructure.

Spontaneously, community development is co-evolutionary with PPT. In the case of R1 (*Anqing village*), relevant content was coded as infrastructure promotion and public service promotion. The village has made efforts to construct a sewage disposal system and community culture room, and promote transportation by hardening roads, adding streetlights, and enhancing other related transportation facilities. In the case of R2 (*Baishuidong village*), relevant content is coded as rural landscape promotion and environment promotion. The village constructed a designated leisure block and has invested more than three million yuan in promoting the rural landscape.

*4.3. Themes of Institutional Capital*

Institutional capital provides people access to the policy-making process and identifies the extent to which people enjoy benefits and achieve better livelihood outcomes [4]. Institutional factors have been proven to be profound factors in China's economic and social development [50]. Case studies reveal that the success of PPT projects requires support from the government, namely devoting resources to improving tourism facilities, employees' competency, and net benefits. Themes of institutional capital have been identified, namely brand effects and institutional effects. Community honor and poverty alleviation mechanisms are ranked as each theme's most popular code category.

Community honor represents the achievement of a poverty alleviation strategy and is highlighted by local and central governments. China has established exemplary norms as a helpful tool for governance [36]. More aid from the government is provided once the village is designated as a model village [37]. According to the results of the coding system, except for *Gangsha*, the rest of the cases have been honored with several designations, such as "National key village of rural tourism", "Chinese beauty village", "Chinese ethnic minority style villages", "National civilization village", etc. Brand designation boosts the exposure of model villages and brings more social and economic capital.

Poverty alleviation mechanisms ensure the success of PPT. The Chinese government supports tourism poverty alleviation [30,32]. According to the results of the coding system, the government's supports for PPT are manifold, including financial support, poverty alleviation through industry, free skill training, cultural construction, etc. For example, in the case of R1 (*Anqing village*), the department of culture and tourism of Guangdong province has funded more than seven million yuan into the poverty alleviation industry, community development, and into cooperatives of the poor. Additionally, the poverty alleviation strategy of PPT in *Anqing* could be summarized as "*Tourism plus agriculture plus poverty alleviation*". Concurrently, "*Company plus cooperatives plus planting base plus household*", "*Brand plus company plus planting base plus the poor*", "*Association plus happy family plus the poor*", and "*Scenic area plus collective plus household*" strategies also have been

applied in various PPT cases. The finding is in accordance with the research of Zhao [63], the implementation pathways of the PPT models are conditional [38].

### 4.4. Themes of Social Capital

Social capital represents the social resources upon which people pursue their livelihood objectives, including the collaboration of networks, social relations, affiliations, and associations [4,40]. China has mobilized all sectors of society to contribute to its poverty alleviation strategy, especially PPT initiatives. Abundant social resources from multiple stakeholders are coordinated to strengthen a community's livelihood. Social resources were recognized as a unique theme based on the coding system, while collaboration networks and social investment acted as the code category.

Social resources are gathered to construct collaboration networks and catch social investment. In the case of R6 (*Dingzhuang village*), relevant content was coded as collaboration networks and social investment. The village has established a series of collaboration networks that work together with e-commerce platforms, social media platforms, and supermarkets to sell grapes and their byproducts. Moreover, *Dingzhuang* advocates rural tourism through collaboration with social media platforms, exhibitions, and neighboring tourism resorts. In addition, *Dingzhuang* convened a professional planning and design company to provide knowledge and skills to promote community development. Social investments such as finance, knowledge, and skills play a significant role in boosting community livelihood.

### 4.5. Themes of Natural Capital

Natural capital constitutes the natural resource stocks and environmental services derived from a community's livelihood and lays the foundation of community development [4,40]. In China, the most poverty-stricken areas are locked in adverse circumstances. Along with the implementation of PPT, community infrastructures are improved, and adverse natural conditions are transformed into natural capital, enhancing the sustainability of community livelihoods. The coding system identified geographic advantage as a unique theme, while tourism resources and location conditions were the code categories. In the case of R10 (*Huaguoshan village*), the government promoted the location conditions by promoting tourism infrastructure, such as road construction. Additionally, natural resources in *Huaguoshan* are transformed into tourism resources since the PPT models are applied.

### 4.6. Themes of Livelihood Outcomes

Livelihood outcomes result from a sustainable community livelihood, reflecting the benefits and promotions that occur under multiple livelihood strategies. PPT has been proven to be an effective tool for realizing sustainable livelihood outcomes, and the poor benefit from tourism livelihoods in the long term [4]. Sustainable livelihood outcomes have multiple objectives. The case studies indicate that poverty alleviation, sustainable tourism livelihoods, and quality of life were conclusive themes, evidenced by tourism's benefits, ability to eradicate poverty, and success in generating improvement in living conditions, respectively.

Livelihood outcomes are highlighted through government advocacy and achievements of PPT are outlined in the second section of each case. Government narratives on R7 (*Gangsha village*) indicate that the village has eradicated poverty in 2019 using PPT. Furthermore, revenue from tourism services has increased to more than three million yuan, and living conditions and quality of life have been improved. In summary, the livelihood outcomes are in accord with the criteria of the rural revitalization strategy in China.

## 5. Discussions

This study tries to uncover the secrets of PPT in China from a government perspective. A sustainable livelihood framework and thematic analysis are combined to decode these exemplars in China. By examining PPT cases, this study has revealed the livelihood effects and several key experiences implied by PPT, as discussed further below.

Evidence from PPT studies conducted in developing countries reveals that unsustainable and short-term investment, loose systems, inadequate participation, and uneven distribution of benefits are the primary barriers to its success [20,23,64,65]. Most of the weaknesses of PPT are rooted in neoliberalism which usually shifts the responsibility to the poor themselves [24]. In contrast, the Chinese government takes a state capitalism strategy and dominates the implementation of PPT in adaptive ways [5,12,17]. Under rural revitalization policies, investment in rural tourism development is long-term, guaranteeing the improvement of public infrastructures, scenic spots, and tourism facilities. Furthermore, PPT exemplars are well supported by the government through finance, technology, and knowledge policies [31,37].

The participation of the poor determines the power of PPT in the realization of SDG1 [32,66]. One-size-fits-all approaches have been abandoned, and a variety of participation models are set up to relieve poverty. For example, land transfer is widely accepted as the poor's major method for participating in PPT [67], and the poor can be trained as tourism employees in PPT as well. Due to the consistent structural inequality of tourism development, a proactive interventionist approach is needed to make the poor benefit from tourism growth [24]. Additionally, community governance is necessary due to the existence of poor populations characterized by weak livelihood capital [17]. It should be noted that elites comprised of CPC members and squires led the participation in the PPT [7,68]. They are qualified with ample institutional, human, social, or economic capital that leads the transformation of community livelihood. Therefore, it is important to identify poor activists who can lead rural communities in PPT initiatives.

Furthermore, the sustainability of PPT is critical for the poor. Research on PPT in Africa and Southeast Asia indicates that a high dependence on foreign companies may marginalize the poor from the benefits generated by tourism [11]. Conversely, China emphasizes the domestic market in spite of its lower consumption capacity [3]. Domestic tourists may be more likely to consume at locally owned establishments that involve more locals, thereby providing greater benefits [17,69]. During the post-pandemic era, place-based rural tourism would attract more domestic tourists [70], providing opportunities for the PPT. China advances PPT by developing abundant tourism products to meet the localization demand.

More importantly, the PPT model mobilizes all livelihood capital to benefit the poor. Limited capital is the most significant obstacle to poverty alleviation [20]. Research has revealed that cooperation between local governments and investors in PPT is the most popular model for transforming a community's livelihoods. The poor are well organized by collective agency and constructed cooperatives, while the enterprises provide opportunities to participate [11]. Under the PPT model, collaboration networks are established to attract the investment of knowledge, personnel, and finance. With contributions from widespread public and government support, the development of PPT in China has been dramatic [71].

To maximize the benefits to livelihoods, alternative strategies are used to improve the resilience of PPT and community livelihood. The cases indicate that the poverty alleviation industries also play a significant role in the consolidation of economic capital. Along with the implementation of PPT, methods of industry integration have been practiced in each case. PPT exemplars have gained marvelous success through the co-evolution of tourism and rural development [61].

It should be noted that the poor need to be targeted by the PPT strategy [17]. In 2013, president Xi Jinping put forward the theory of Targeted Poverty Alleviation, which lays out the fundamental strategy for tackling poverty. In China, a fixed standard has been set, namely that people with a per capita annual net income of less than 2300 yuan would be targeted as the poor. However, mistargeting the poor would weaken the efficiency of PPT.

PPT in China indicates that paradigm shifts in rural tourism are effective and necessary anti-poverty tools [6]. Thus, the grasp of the institutional context is critical for PPT. Institutions determine whether poverty alleviation is an ambitious political strategy bound to be achieved by government strategy. Exemplars certify that the outcomes of PPT are in accordance with

the criteria of the Beautiful Village designation in China. With sound policies, plans, and practices that are used in tourism development, the poor's profit from the PPT is ensured. Based on the transformation of local livelihoods and the engagement of society as a whole, PPT can greatly enhance the positive impacts on the economy and poverty.

## 6. Conclusions

Tourism acts as a tool for poverty alleviation by diversifying rural economic activity and livelihoods [27,41,52]. To make tourism more pro-poor, pro-poor tourism is proposed and practiced globally. In China, as of 2020, PPT has contributed to the achievement of SDG1. China's success story provides new evidence for the broad debate over the poverty alleviation effects of PPT [9]. Exemplary norms can be seen in the traditional governance models in China [36] and the selected PPT cases shown here could shed light on the experience of China's effective poverty alleviation strategies through tourism development.

Understanding the tourism-poverty link is critical for reducing poverty and achieving the SDGs [6,17]. Studies on the tourism-economic growth link have been carried out to answer questions such as 'Does tourism reduce poverty?' [18,29,59], 'Under what conditions is this link likely to be strongest?' [17] and 'Whether tourism benefits the poor more than non-poor people' [23,71]. Therefore, making PPT an effective poverty alleviation tool is an urgent and worthy objective to be accomplished. PPT principles are comprised of participation, a holistic livelihoods approach, a balanced approach, wide application, distribution, flexibility, commercial realism, and cross-disciplinary learning [23]. Clearly, prerequisites for PPT as a useful tool to eliminate poverty are complex and reflect pluralism [3].

In summary, the experience of PPT in China could be concluded as follows. Firstly, PPT is dominated by the government, and institutional capital is a necessary element for addressing community livelihoods. Secondly, PPT needs a well-organized community, and measures such as grassroots CPC construction and community regulations are combined to promote human capital. Thirdly, PPT focus on the all-around participation of the locals, and tourism participation is encouraged because of its low threshold for participation. Lastly, PPT requires the integration of social resources inside and outside of the community, while keeping the balance between stakeholders.

Nonetheless, some limitations are hard to avoid and should be explored further. Government perspectives increase the limitation of our findings, and the resilience and sustainability of PPT may be weakened under the burst of COVID-19. Thus, a specific field study would be helpful. Moreover, integration-oriented industries are a unique type selected because they are characterized by multiple livelihood strategies. A comprehensive study of all cases needs to be conducted in future research to provide in-depth knowledge of PPT.

**Author Contributions:** Conceptualization and paper writing, Z.W.; Project administration and review, F.D.; All authors have read and agreed to the published version of the manuscript.

**Funding:** This research was funded by Shenyang Philosophy and Social Sciences (SQ202001L) and National Natural Science Foundation of China (42101294).

**Data Availability Statement:** Not applicable.

**Acknowledgments:** The author sincerely thanks for editors' work and advisement of anonymous reviewers.

**Conflicts of Interest:** The author declares no conflict of interest.

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
