# Peer review of "Experience of Pro-Poor Tourism (PPT) in China: A Sustainable Livelihood Perspective"

_sustainability, doi:10.3390/su142114399_

Round 1

Reviewer 1 Report

This research attempts to provide insights on PPT in China, esp Liaoning Province. A very timely study in assessing the effectiveness of policy. Congratulations to the author.

I'd encourage the author to add more critical comments in the research question: what is the potential negative impact of the current policy? Your research should also be engaged in addressing the problems and improvement of the situation.

You may also address this issue in the discussions.

Reviewer 2 Report

Comments

sustainability-1960169

Experience of Pro-poor tourism (PPT) in China: A sustainable livelihood perspective

This study examines the government PPT scheme with a sustainable livelihood framework, uncovering the knacks of the PPT in poverty alleviation. With thematic analysis, 18 cases are systematically coded, and several findings are discovered. Rural development is co-evolutionary with the PPT, while livelihood capitals change correspondingly. Specifically, human capital is mentioned as the top priority, followed by economic capital, institutional capital, social capital, and natural capital. Analysis of the cases indicates that livelihood capitals comprise multiple themes, and a variety of livelihood strategies are applied conditionally. Besides, livelihood outcomes are in accord with the criteria of Beautiful Village, characterized by good quality of life. In summary, the success of the PPT in China is a comprehensive project, contributed by government-led model, well-organized community system, effective community participation, and whole of society synergy. The study demonstrates that a paradigm shift has practiced in China’s PPT model and sheds light on tourism development in poverty alleviation globally.

      1.       This is an interesting paper.

2.       The authors should make clear the contribution of your article to the international literature in the introduction.

3.       The authors enrich their references and literature review in two parts: a) to analyze the relationship between poverty and tourism (Croes, 2014); and b) to analyze the determinants variables of tourism (Environmental Sustainability: Szolnoki and Tafel, 2022, health quality: Konstantakopoulou, 2022). These references are essential.

4.       The authors should analyze the Thematic analysis of the PPT model.

     5.       The authors should analyze the Thematic analysis of the PPT model. In addition, the authors should state their reasons for selecting this analysis.

6.       The authors should work more and present the empirical part of the article better.

7.       The authors should put more effort and thoroughly discuss point estimates, estimated effects, and the intuition behind the results backed up by the literature.

  References

 Croes, R., 2014. The role of tourism in poverty reduction: an empirical assessment. Tourism Economics, 20 (2), 207-226.

Konstantakopoulou, I., 2022. Does health quality affect tourism? Evidence from system GMM estimates. Economic Analysis and Policy, 73, 425-440.

Szolnoki, G.; Tafel, M., 2022. Environmental Sustainability and Tourism—The Importance of Organic Wine Production for Wine Tourism in Germany. Sustainability14, 11831. https://doi.org/10.3390/su141911831

Reviewer 3 Report

Thank you for the opportunity of reading and reviewing your interesting manuscript. The topic is important by itself but particularly for China, given the expansion of tourism in the world on one side, and the importance of using all available tools for fighting against the poverty. Pro-poor tourism is in this context a way for inclusion of larger segments of population in activties for leisure and recreation and also to develop and enhance tourism sector.

Regarding the structure of the paper, I appreciate it and mu only recommendation is to separate the Discussion section by the Conclusions section. In the Discussion section your should present and comment all the results and emphasize their importance. In the Conclusions section you should present how you achieved the goals and respond to the research questions, which are the theoretical and practical/policy implications of your findings, and to highlight the limitations and possible further investigations. The Discussion can be definitely enhanced.

I am suggestions also to present in the first section which is the research gap you want to fill in and which are the research questions for discussion.

Some minor comments: check all the acronyms, I haven t found what is CMTC? Is the same as MCTC?

Good luck!

Reviewer 4 Report

The manuscript investigates the government PPT scheme with a sustainable livelihood framework, uncovering the knacks of the PPT in poverty alleviation. The manuscript is of good quality and the goals are clearly articulated and the methods are sound. I recommend that this manuscript be accepted after the author undertake minor revisions. Below are specific comments for the author.

(1) The DFID framework is the most used one, which contains five dimensions, Natural capital, Physical capital, Financial capital, Human capital, and Social capital. But previous research also claimed that Cultural capital is also an important factor. Why is this dimension not used in this article?

(2) More information is needed to better justify the theoretical contributions and practical significance in the Discussion section.

(3) Language needs to be polished a bit.

Round 2

Reviewer 2 Report

The authors enrich their references and literature review in two parts: a) to analyze the relationship between poverty and tourism (Croes, 2014); and b) to analyze the determinants variables of tourism (Environmental Sustainability: Szolnoki and Tafel, 2022, health quality: Konstantakopoulou, 2022). These references are essential.

 References

 Croes, R., 2014. The role of tourism in poverty reduction: an empirical assessment. Tourism Economics, 20 (2), 207-226.

Konstantakopoulou, I., 2022. Does health quality affect tourism? Evidence from system GMM estimates. Economic Analysis and Policy, 73, 425-440.

Szolnoki, G.; Tafel, M., 2022. Environmental Sustainability and Tourism—The Importance of Organic Wine Production for Wine Tourism in Germany. Sustainability14, 11831. https://doi.org/10.3390/su141911831
